# Meta-Gradient Search Control: A Method for Improving the Efficiency of Dyna-style Planning

**Bradley Burega**[*]
*University of Alberta, Amii*

*burega@ualberta.ca*

**John D. Martin**[*]
*Intel Labs, University of Alberta*

*john.martin@intel.com*

**Luke Kapeluck**
*University of Alberta, Amii*

*kapeluck@ualberta.ca*

**Michael Bowling**
*University of Alberta, Amii*

*bowling@ualberta.ca*

## Abstract

We study how a Reinforcement Learning (RL) system can remain sample-efficient when learning from an imperfect model of the environment. This is particularly challenging when the learning system is resource-constrained and in continual settings, where the environment dynamics change. To address these challenges, our paper introduces an online, meta-gradient algorithm that tunes a probability with which states are queried during Dyna-style planning. Our study compares the aggregate, empirical performance of this meta-gradient method to baselines that employ conventional sampling strategies. Results indicate that our method improves efficiency of the planning process, which, as a consequence, improves the sample-efficiency of the overall learning process. On the whole, we observe that our meta-learned solutions avoid several pathologies of conventional planning approaches, such as sampling inaccurate transitions and those that stall credit assignment. We believe these findings could prove useful, in future work, for designing model-based RL systems at scale.

## 1 Introduction

Despite the many successes of Reinforcement Learning (RL) (Mnih et al., 2015; Silver et al., 2016; Wurman et al., 2022), sample-efficiency remains a key issue preventing its further adoption in new technologies and in the science of intelligence. Model-based approaches offer a promising solution to the issue; these methods boost sample-efficiency by generating additional, simulated learning experiences from an internal environment model (Deisenroth & Rasmussen, 2011; Chua et al., 2018; Saleh et al., 2022; Hafner et al., 2023). The process of learning from simulated experience is generally known as *planning*.

To a large extent, the effectiveness of a model-based approach depends on the efficiency of its planning process. To illustrate this point, consider two systems: one that forward-simulates different ways to achieve a goal, and another that simulates unlikely and irrelevant scenarios. Clearly, the system that plans with goal-relevant experience will be able to achieve greater performance given the same amount of experience, and hence greater sample-efficiency than the alternative. Furthermore, one can surmise that improved planning-efficiency usually translates to improved sample-efficiency.

Prior work has confirmed this intuition. Early work on Prioritized Sweeping highlighted the importance of planning from states where value estimates are inaccurate (Moore & Atkeson, 1993). This insight led to a line of efficient model-based algorithms (Peng & Williams, 1993; Andre et al., 1997; Wingate et al.,

---

[*]Equal contribution.

2005); however, these only performed well on a niche class of tabular domains. In another line of work, researchers demonstrated how planning-efficiency is tied to imperfections of an environment model (Talvitie, 2017; Jafferjee et al., 2020; Abbas et al., 2020). These studies emphasized the importance of planning from states where the environment model is trustworthy. Interestingly, however, these prior works imposed sampling preferences with fixed strategies. The process of choosing samples with which to query a model has been called *search control* (Sutton & Barto, 2018).

Our work studies the problem of *learning* to perform search control, which has thus far received little attention in RL research. We focus on Dyna-style algorithms, known for interleaving learning experiences from a model and from the environment for credit assignment (Sutton, 1991; Sutton et al., 2012; Feinberg et al., 2018; Hafner et al., 2019; 2023). We propose a meta-learning (Beck et al., 2023) algorithm that evaluates model queries based on the samples' ability to improve efficiency of the downstream planning process. Operationally, our algorithm draws samples from a distribution over initial states and modulates the associated probabilities with meta-gradients (Xu et al., 2018; Flennerhag et al., 2022). We conduct an empirical study in two non-stationary, stochastic domains; the results demonstrate our algorithm's superior sample-efficiency relative to baselines that employ fixed search control strategies.

## 2 Problem Setting

This work addresses RL problems where model-based approaches are both relevant and necessary. In such settings, an agent may interact with a relatively large, complex world which can appear non-stationary. The agent's interactions are based on finite sets of actions $\mathcal{A}$ and observations $\mathcal{O}$; where, at every moment in time $t \in \mathbb{N}$, the agent takes an action $a_t \in \mathcal{A}$ and subsequently observes the outcome, $o_{t+1} \in \mathcal{O}$, and a scalar reward $r_{t+1}$. A sequence of interactions is referred to as a history, $h = a_1, o_1, a_2, o_2, \cdots$, with length-$n$ histories coming from the set $\mathcal{H}_n \triangleq (\mathcal{A} \times \mathcal{O})^n$, and all histories from $\mathcal{H} \triangleq \bigcup_{n=1}^{\infty} \mathcal{H}_n$. To model non-stationarity, the agent is assumed to observe samples from a distribution conditioned on the current history and action, denoted $e \colon \mathcal{H} \times \mathcal{A} \to \Delta(\mathcal{O} \times \mathbb{R})$. Furthermore, as a matter of methodological convenience, in this study, agents interact through episodic experiences[1].

The goal is to learn a policy, $\pi \colon \mathcal{H} \to \Delta(\mathcal{A})$, that maximizes the expected sum of future discounted rewards. For a given discount factor $\gamma \in [0, 1)$, the action-value, $q^\pi(h, a)$, reflects the current utility of taking action $a$ from the history $h$ and following $\pi$ for all timesteps thereafter:

$$q^\pi(h, a) \triangleq \mathbf{E}[R_{t+1} + \gamma R_{t+2} + \gamma^2 R_{t+3} + \cdots | H_t = h, A_t = a, \pi, e].$$

In many settings, it is common for the agent to follow an $\epsilon$-greedy policy; this selects uniform-random actions with probability $\epsilon$ and otherwise selects actions that maximize the current action-value.

As it is generally impractical for the agent to use the full history when computing values or selecting actions, the agent is assumed to maintain a finite-size approximation, known as its *internal state* $s \in \mathcal{S}$; at any given moment, this provides context for the agent's present circumstances in the environment[2]. Following prior work (Dong et al., 2022; Sutton et al., 2022; Abel et al., 2023), we define the internal state recursively, as $s_{t+1} \triangleq f(s_t, a_t, o_{t+1})$, for all timesteps $t$, and $f \colon \mathcal{S} \times \mathcal{A} \times \mathcal{O} \to \mathcal{S}$ taken as the state update function. Henceforth, we use "state" and "internal state" synonymously.

Additionally, the agent forms an approximate value function $\hat{q}(s, a; \boldsymbol{\theta}) \approx q(h, a)$, with a vector of real-valued parameters $\boldsymbol{\theta}$. In large-scale settings, both $s$ and $\hat{q}$ are typically composed in a single, deep neural network; the internal state, in these cases, can be viewed as the output from the penultimate layer, and value estimates are the output from the final layer (Mnih et al., 2015).

---

[1]As our algorithm does not critically depend on episodic structure, we believe that it could be applied to non-episodic settings without difficulty.

[2]In fully-observable settings, the current observation is often identical to the internal state.

## 2.1 Learning from a Model

Model-based RL systems are characterized by their use of an internal environment model, $m$. Typically a model generates experiences, in the form of transition tuples $(\tilde{s}, \tilde{a}, \tilde{r}, \tilde{s}')$, and an agent uses this data to inform policy updates. For instance, AlphaGo (Silver et al., 2016) uses a model for action evaluation and selection, in a process sometimes called "decision-time-planning." In contrast, Dyna algorithms (Sutton, 1991) use models for credit assignment; given a state and action, $\tilde{s}, \tilde{a}$, the algorithms treat model outputs, $\tilde{s}', \tilde{r} \sim m(\tilde{s}, \tilde{a})$, as if they came directly from the environment—using them to update the approximate value function. The psuedocode for this algorithm is provided in Algorithm 1. As part of planning, the agent employs a learning rule to update its value parameters, $\boldsymbol{\theta}$ (e.g., $Q$-Learning (Watkins & Dayan, 1992)). In this work, a model contains two components, $m = (p, r)$; the first, $p \colon \mathcal{S} \times \mathcal{A} \to \Delta(\mathcal{S})$, predicts future observations, and the second, $r \colon \mathcal{S} \times \mathcal{A} \to \Delta(\mathbb{R})$, predicts rewards.

---

**Algorithm 1** Dyna

---

1: Obtain initial state, $s_1$.
2: **for** $t = 1, 2, 3, \cdots$ **do**
3:     Take action $a_t$ from $s_t$ then obtain $s_{t+1}$ and $r_{t+1}$.
4:     $m \leftarrow \text{UpdateModel}(m, s_t, a_t, s_{t+1}, r_{t+1})$
5:     # Perform a direct update.
6:     $\boldsymbol{\theta} \leftarrow \text{UpdateValue}(\boldsymbol{\theta}, s_t, a_t, s_{t+1}, r_{t+1})$
7:     # Perform $k$ planning updates.
8:     **for** $1, \cdots, k$ **do**
9:         Query model $m$ from $\tilde{s}, \tilde{a}$ to obtain $\tilde{s}', \tilde{r}$.
10:        $\boldsymbol{\theta} \leftarrow \text{UpdateValue}(\boldsymbol{\theta}, \tilde{s}, \tilde{a}, \tilde{s}', \tilde{r})$

---

## 2.2 Learning a Model

Model-based systems usually start with little knowledge of their environment. In such cases, they must learn their model from data gathered during interaction. Systems can use non-parametric models, such as empirical distributions or replay buffers, to represent the unknown distributions $p$ and $r$. Alternatively, systems can use parametric models (e.g., tables of counts or neural networks) to compute transition likelihoods or mimic the generative nature of sampling distributions. Many systems train their models to minimize a reconstruction error (Hafner et al., 2019; 2023); however, alternative formulations are being explored in recent work (Silver et al., 2017; Schrittwieser et al., 2020; Saleh et al., 2022). Non-stationarity has been modeled with change-point detectors (Alegre et al., 2021), variational inference (Xie et al., 2021), and other structural changes (Chen et al., 2022).

## 2.3 Querying a Model (Search Control)

Search control addresses the question of how to query a model; that is, how to determine the ~~initial~~ state and action on which $m$ conditions. Preferences, regarding which scenarios to prioritize, are defined by a search control strategy, with a particular strategy defined by a joint distribution $p_1 \in \Delta(\mathcal{S} \times \mathcal{A})$. Specifically, a strategy, $p_1$, imposes preferences through its probabilities over state-action pairs, because states with higher probability mass are more likely to be selected for planning updates. Furthermore, every strategy factors into two distributions: $p_1(\tilde{s}, \tilde{a}) = \tilde{\pi}(\tilde{a}|\tilde{s})d(\tilde{s})$; the first is a one-step policy, $\tilde{\pi} \colon \mathcal{S} \to \Delta(\mathcal{A})$, that conditions on samples from ~~initial~~ search control distribution $d \in \Delta(\mathcal{S})$. In our work, $\tilde{\pi}$ is defined as the behavior policy, $\tilde{\pi} = \pi$, and the ~~initial~~ search control distribution is parameterized by a real-valued vector $\boldsymbol{\eta}$. To construct a query, the agent first draws a state $\tilde{s} \sim d(\cdot; \boldsymbol{\eta})$ then draws an action $\tilde{a} \sim \tilde{\pi}(\cdot|\tilde{s})$. Interestingly, if the probabilities on each state are non-zero, and $\mathcal{S}$ and $\mathcal{A}$ are finite, then value iteration is still guaranteed to converge under typical conditions on the step-size (Tsitsiklis, 1994; Bertsekas, 2015).

### 2.4 Meta-learning

Meta-learning algorithms optimize the hyperparameters or procedure of a base learning algorithm (Schmidhuber, 1987; Hinton & Plaut, 1987; Bengio et al., 1990; Beck et al., 2023). Their objective can be to improve performance online (Xu et al., 2018; Zahavy et al., 2020; Flennerhag et al., 2022) or over a distribution of goals (Finn et al., 2017). In the context of RL, meta-gradients (Xu et al., 2018; Flennerhag et al., 2022) have been used to tune parameters such as the step-size, discount factor, and eligibility trace parameter. Our work takes the search-control distribution, $d$, as the candidate for meta-optimization and specifically applies a gradient-based algorithm to tune the parameters $\boldsymbol{\eta}$.

## 3 Meta Gradient Search Control

In this section, we introduce an algorithm for learning to perform search control. Our algorithm, Meta Gradient Search Control (MGSC), is capable of adjusting its sampling distribution over model states to improve the efficiency of its planning process. In what follows, we derive MGSC's meta-loss and describe how it can boost the efficiency of Dyna-style planning[3].

### 3.1 The Meta-Loss

The MGSC meta-loss reflects a general desire to maximize planning-efficiency. Although the term "efficiency" can take on many meanings, here, we use it to describe the degree to which a value estimate, $\hat{q}(s, a; \boldsymbol{\theta})$, contracts toward its optimal fixed point, $\hat{q}(s, a; \boldsymbol{\theta}^*)$ given a fixed number of planning updates. To illustrate this concept, consider a scenario where the learning system evaluates the efficiency of a single query, $\tilde{s} \sim d(\cdot; \boldsymbol{\eta})$; the learner asks: "How close did my planning update, from $\tilde{s}$, bring me to the optimal parameters, $\boldsymbol{\theta}^*$?" Pretend the optimal parameters are available. In addition, denote the updated parameters (i.e. post-planning) by $\bar{\boldsymbol{\theta}}$. Closeness can then be measured in terms of squared Euclidean error: $||\boldsymbol{\theta}^* - \bar{\boldsymbol{\theta}}||_2^2$.

In reality, the optimal parameters are not available, and the post-planning parameters depend on the search control strategy, $\bar{\boldsymbol{\theta}}(\boldsymbol{\eta})$. We address the first issue with an approximation: $\boldsymbol{\theta}^* \approx \hat{\boldsymbol{\theta}}$. The approximate targets, $\hat{\boldsymbol{\theta}}$, are computed by performing an additional update to the post-planning parameters, using experience obtained directly from the environment. In formal terms, let a semi-gradient $Q$-Learning update to $\boldsymbol{\theta}$, from the transition $(s, a, r, s')$, and with a step-size $\alpha \in \mathbb{R}_+$ be

$$\Delta(s, a, r, s'; \boldsymbol{\theta}) \triangleq [r + \gamma \max_{a' \in \mathcal{A}} \hat{q}(s', a'; \boldsymbol{\theta}) - \hat{q}(s, a; \boldsymbol{\theta})] \nabla_{\boldsymbol{\theta}} \hat{q}(s, a; \boldsymbol{\theta}).$$

Then, the approximate targets are defined as $\hat{\boldsymbol{\theta}}(\boldsymbol{\eta}) \triangleq \bar{\boldsymbol{\theta}}(\boldsymbol{\eta}) + \alpha \Delta(s, a, r, s', \bar{\boldsymbol{\theta}}(\boldsymbol{\eta}))$. To encourage optimization stability, we suppress the target's dependence on $\boldsymbol{\eta}$ with a stop-gradient and, with an abuse of notation, write $[\![\hat{\boldsymbol{\theta}}]\!] = [\![\hat{\boldsymbol{\theta}}(\boldsymbol{\eta})]\!]$. The post-planning parameters are computed with an expected update, given $\tilde{s} \sim d(\cdot; \boldsymbol{\eta}), \tilde{a} \sim \pi(\cdot|\tilde{s})$, and $\tilde{s}', \tilde{r} \sim m(\tilde{s}, \tilde{a})$:

$$\bar{\boldsymbol{\theta}}(\boldsymbol{\eta}) \triangleq \boldsymbol{\theta} + \alpha \sum_{\tilde{s}, \tilde{a}} \pi(\tilde{a}|\tilde{s}) d(\tilde{s}; \boldsymbol{\eta}) \Delta(\tilde{s}, \tilde{a}, \tilde{r}, \tilde{s}'; \boldsymbol{\theta}).$$

This is intended to encourage equal credit assignment among all the initial states and actions. After putting the preceding definitions together, we obtain the MGSC meta-loss. Minimizing this meta-loss improves planning-efficiency by design:

$$\mathcal{L}(\boldsymbol{\eta}) \triangleq ||[\![\hat{\boldsymbol{\theta}}]\!] - \bar{\boldsymbol{\theta}}(\boldsymbol{\eta})||_2^2. \tag{1}$$

### 3.2 The Search Control Strategy

Recall a search control strategy is given by the distributions $\tilde{\pi}$ and $d$. In our work, $\tilde{\pi}$ is fixed to the behavior policy, so $d$ is learned by minimizing (1). We represent $d$ as a softmax distribution and encode a logit for

---

[3]Although our paper focuses on Dyna, we believe the MGSC methodology is more generally applicable.

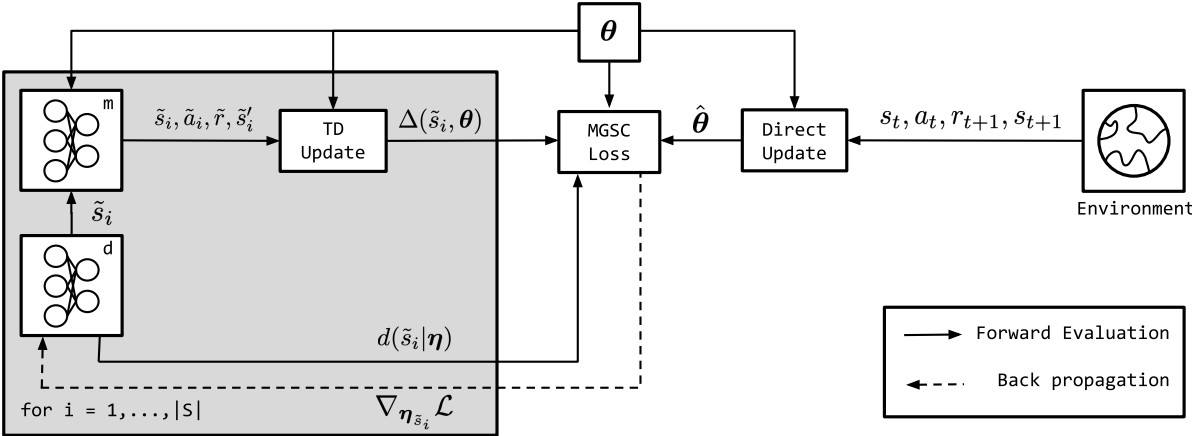

Figure 1: System diagram of training with Meta Gradient Search Control. The gray box denotes replication over the index $i$. The initial value parameters $\boldsymbol{\theta}$ are used for computing actions in the model $m$, the update operations, and in the MGSC loss.

each state with a component of $\boldsymbol{\eta}$; each is denoted $\boldsymbol{\eta}_s$, for all $s \in \mathcal{S}$:

$$d(s; \boldsymbol{\eta}) \triangleq \frac{e^{\boldsymbol{\eta}_s}}{\sum_{i=1}^{|\mathcal{S}|} e^{\boldsymbol{\eta}_s}} = \mathbb{P}(s|\boldsymbol{\eta}).$$

When the number of internal states is large, it may be possible to fix the number of logits, $n \in \mathbb{N}$, and use a neural network to output them as a function of the state, replacing the $\boldsymbol{\eta}_s$ with $\boldsymbol{\eta}_i(s)$ above, for all $i = 1, \cdots, n$. Alternatively, there are several other representations available for distributions, including random networks, normalizing flows (Papamakarios et al., 2021), variational auto-encoding (Kingma & Welling, 2013), and probabilistic graphical models (Papamakarios et al., 2021; Kingma & Welling, 2013). We leave it to future work to explore these possibilities.

### 3.3 Meta Gradient Search Control in Dyna

Algorithm 2 outlines the MGSC procedure for Dyna. The algorithm assumes the use of an $\epsilon$-greedy behavior policy. Furthermore, the algorithm performs online updates to the value function using semi-gradient $Q$-Learning updates, which support non-linear function approximation. The MGSC loss (1) is minimized using Adam (Kingma & Ba, 2014); gradients are back-propagated through $\bar{\boldsymbol{\theta}}$ and into the distribution $d(\boldsymbol{\eta})$ (see Figure 1 for an illustration of this computation).

**Algorithm 2** Meta-Gradient Search Control in Dyna

---

1: Obtain initial state, $s_1$.
2: **for** $t = 1, 2, 3, \cdots$ **do**
3:     Take $\epsilon$-greedy action $a_t$ from $s_t$ then obtain $s_{t+1}$ and $r_{t+1}$.
4:     $m \leftarrow \text{UpdateModel}(m, s_t, a_t, s_{t+1}, r_{t+1})$
5:     # Perform a direct update.
6:     $\boldsymbol{\theta} \leftarrow \boldsymbol{\theta} + \alpha[r + \gamma \max_{a'} \hat{q}(s', a'; \boldsymbol{\theta}) - \hat{q}(s, a; \boldsymbol{\theta})]\nabla_{\boldsymbol{\theta}} \hat{q}(s, a; \boldsymbol{\theta})$
7:     # Perform $k$ planning updates.
8:     **for** $1, \cdots, k$ **do**
9:         Take $\epsilon$-greedy $\tilde{a}$ from $\tilde{s} \sim d(\cdot; \boldsymbol{\eta})$.
10:        $\tilde{s}', \tilde{r} \sim m(\tilde{s}, \tilde{a})$.
11:        $\boldsymbol{\theta} \leftarrow \boldsymbol{\theta} + \alpha[\tilde{r} + \gamma \max_{\tilde{a}'} \hat{q}(\tilde{s}', \tilde{a}'; \boldsymbol{\theta}) - \hat{q}(\tilde{s}, \tilde{a}; \boldsymbol{\theta})]\nabla_{\boldsymbol{\theta}} \hat{q}(\tilde{s}, \tilde{a}; \boldsymbol{\theta})$
12:     # Construct post-planning parameters.
13:     $\bar{\boldsymbol{\theta}}(\boldsymbol{\eta}) \leftarrow \boldsymbol{\theta} + \alpha \sum_{\tilde{s}, \tilde{a}} \pi(\tilde{a}|\tilde{s}) d(\tilde{s}; \boldsymbol{\eta}) \Delta(\tilde{s}, \tilde{a}, \tilde{r}, \tilde{s}', \boldsymbol{\theta})$
14:     # Construct approximate target parameters.
15:     $\hat{\boldsymbol{\theta}} \leftarrow \bar{\boldsymbol{\theta}} + \alpha \Delta(s, a, r, s', \bar{\boldsymbol{\theta}})$
16:     Update $\boldsymbol{\eta}$ with Adam on the MGSC meta-loss using $\hat{\boldsymbol{\theta}}$ and $\bar{\boldsymbol{\theta}}$ (1).

---

## 4 Empirical Analysis

This section establishes supporting evidence for the claim that MGSC can improve sample-efficiency of model-based RL systems. Evidence comes in the form of empirical results, with data gathered in multiple non-stationary domains. Comparisons are made with multiple systems, based on the pseudocode in Algorithm 2, using total-reward over a fixed number of timesteps as a measure of sample-efficiency. Using total reward as our evaluation metric allows us to measure the level of performance each agent achieves given the same amount of interaction with the environment. For complete details regarding our methodology, please refer to the Appendix.

Our study begins in a modest setting, where the factors of variation are tightly controlled. With each new experiment, the learning problem becomes increasingly difficult. First, we control for the effects of learning an environment model, simultaneously, with a search control strategy; we hold the model fixed at an approximate, limit state. Next, the search control strategy is learned with the model concurrently. In the final set of experiments, we enlarge the domain, providing a more challenging environment with more states. In each experiment, we find that MGSC improves the sample-efficiency of the model-based system.

### 4.1 TMaze Experiments

The TMaze is a stochastic gridworld, inspired by early animal-learning experiments from Bush & Mosteller (1953). In our experiments, the environment contains two terminal states; one rewards the agent with a bonus of +1 and the other provides zero reward. The goal location is swapped every 600 episodes—making this environment non-stationary. Appendix A.1 describes the environment in more detail.

We consider three baseline algorithms. The first is a model-free algorithm ($Q$-Learning); its performance sets a lower limit on the model-based algorithms. One model-based algorithm (Uniform) queries initial states with a fixed, uniform distribution: $d = \mathcal{U}(\mathcal{S})$. The other model-based algorithm (Avoid Terminal) uses privileged information about the environment to define its search control strategy; namely, it biases sampling towards states whose values change when the goal swaps and biases sampling away from states where the model is erroneous. Figure 10 in Appendix A.2 provides visualizations of these distributions.

#### 4.1.1 TMaze: Fixed Model

In this experiment, each model-based algorithm is given the same, fixed, imperfect model of the TMaze. The model is a stationary approximation of the true dynamics; it matches the environment in most cases, except at the terminal transitions. At these locations, the model ignores goal switches and, instead, outputs

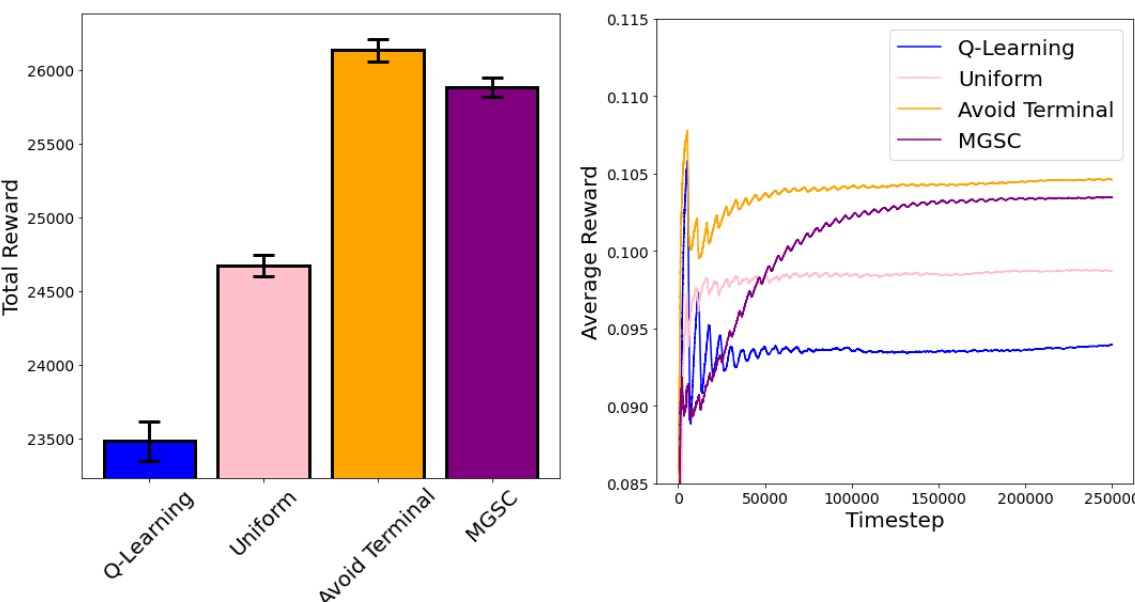

Figure 2: **TMaze Fixed Model Performance**: (a) The total reward reflects the sample-efficiency of each learning algorithm. Error bars denote the 95% confidence interval over 30 seeds. (b) The average reward shows how learning performance varies through time and how each system copes with non-stationarity.

rewards of one or zero with equal probability, thus matching the empirical distribution of observed rewards for these transitions in the limit of experience.

We takeaway several points from the plots in Figure 2. Clearly model-based algorithms are well-suited to this domain, since *Q*-Learning achieves the lowest observed performance. Of the model-based algorithms, Uniform accumulates the least amount of total reward; it performs erroneous and redundant updates with higher frequency, thus suppressing its planning-efficiency. Avoid Terminal, on the other hand, achieves the greatest performance; it makes good use of its planning updates by avoiding terminal transitions and biasing samples toward states where value estimates are inaccurate. MGSC achieves a close-second to Avoid Terminal and, more importantly, outperforms Uniform. Similarly, the average reward of MGSC is well above that of Uniform. This result signifies MGSC's ability to improve sample-efficiency without privileged knowledge of the domain.

The distribution MGSC learns (Figure 3) has several key features; it avoids states where the model is inaccurate (i.e. terminals) and updates are redundant (i.e. the vertical hallway), and it places more probability on states that need updates between goal switches (i.e. the horizontal hallway).

### 4.1.2 TMaze: Learned Model

In this experiment, the environment model is learned alongside the policy. Now the question becomes: can MGSC improve sample-efficiency when the model is flawed and continually updates. The methodology from the previous experiment is repeated.

The learned model is based on counts of observed rewards at each transition. Counts define an empirical distribution, from which the agent samples while planning. Notice this model is a stationary approximation of the TMaze dynamics. And in the limit, the model behaves identically to the fixed model from the previous experiment.

Conclusions drawn from Figure 4 are consistent with the previous experiment. When learning an environment model, MGSC achieves improved performance relative to the baseline algorithms; it now exceeds the performance of Avoid Terminal. Overall, the total reward is lower than it is with a fixed model; this reflects the sample cost to learn a model. The average reward plot shows that MGSC becomes persistently efficient,

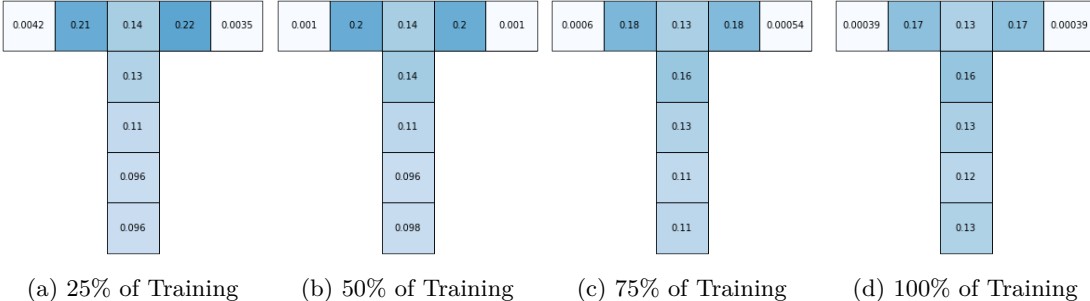

(a) 25% of Training     (b) 50% of Training     (c) 75% of Training     (d) 100% of Training

Figure 3: **TMaze Fixed Model Solution**: the evolution of MGSC's learned search control distribution during the fixed model experiment. Each figure shows MGSC's learned search control distribution, represented by the parameter vector $\boldsymbol{\eta}$, at a different point during training. The number overlaying each state represents the probability that the corresponding state of MGSC's TMaze model will be sampled during a planning step. Darker colors indicate greater probability mass. The distribution is initialized uniformly, but as shown in the Figure, by 25% of training, MGSC's meta-gradient updates have shifted probability away from the states adjacent to terminal states. As the model's terminal states produce erroneous rewards, MGSC learns to avoid planning using them.

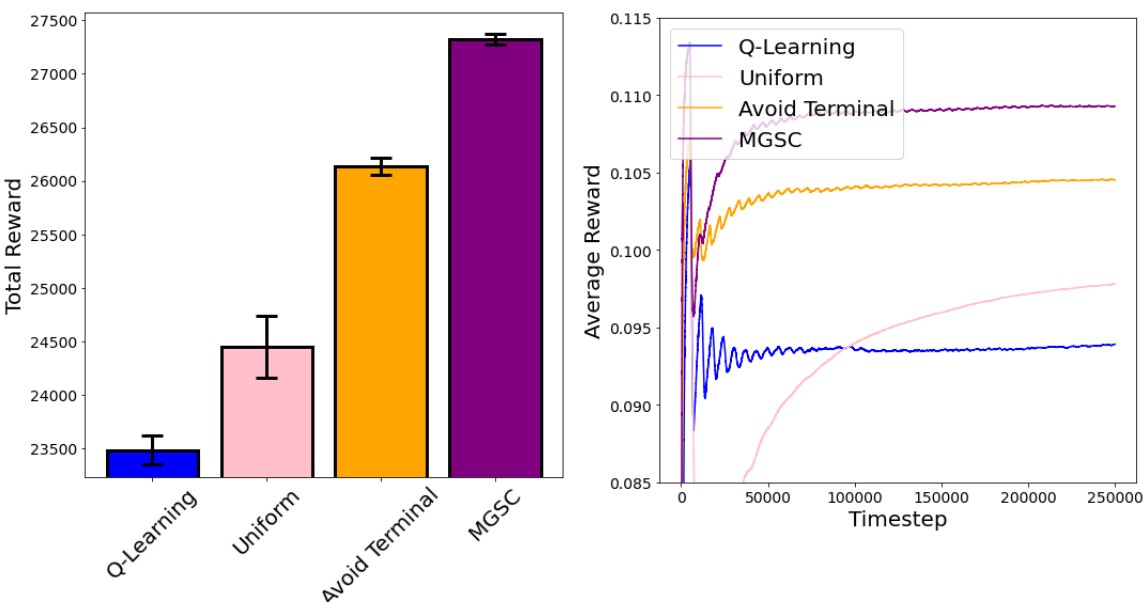

Figure 4: **TMaze Learned Model Performance**: (a) The total reward accumulated by each agent over the course of training. Error bars denote the 95% confidence interval. (b) The average reward accumulated during training for each agent.

and achieves greater average reward than Uniform and Avoid Terminal. The bottomline here is that MGSC achieves the greatest total reward given the amount of experience, demonstrating that it has the highest sample-efficiency.

The distribution MGSC learns resembles its solution from the previous experiment (Figure 5). It's again notable that MGSC concentrates proability away from states with erroneous transitions under the learned model. Although, in this case MGSC concentrates greater probability on the starting state of the TMaze.

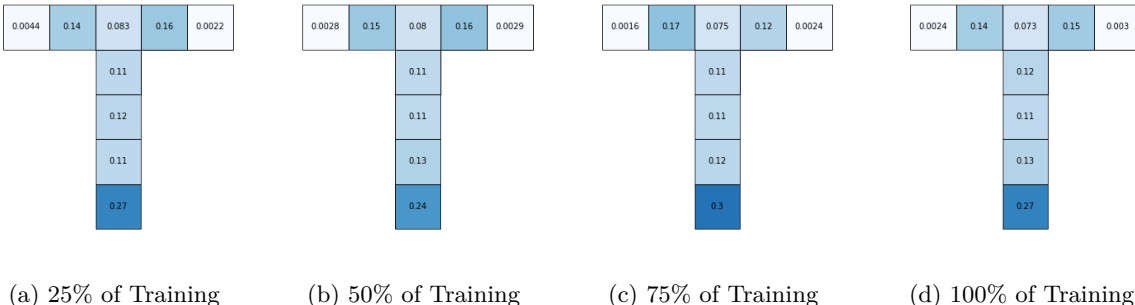

(a) 25% of Training      (b) 50% of Training      (c) 75% of Training      (d) 100% of Training

Figure 5: **TMaze Learned Model Solution**: the evolution of MGSC's learned search control distribution during the learned model experiment. Just as in Figure 3, each figure shows MGSC's learned search control distribution, represented by the parameter vector $\boldsymbol{\eta}$, at a different point during training. The number overlaying each state represents the probability that the corresponding state of MGSC's TMaze model will be sampled during a planning step. Again, the distribution is initialized uniformly, but MGSC's meta-gradient updates adjust the placement of probability mass. In this case, shifting probability mass away from the states adjacent to the terminal state, but also moving greater mass onto the starting state.

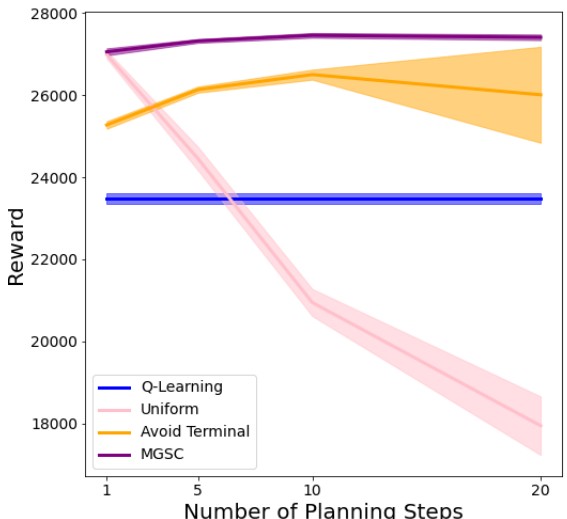

Figure 6: **TMaze Robustness to Imperfections**: A comparison of the total reward accumulated by each agent as the amount of planning is varied. Note that $Q$-Learning does not perform any planning but is included as a baseline. Bold lines indicate averages over all random seeds while shaded regions indicate 95% confidence intervals.

**Robustness to Imperfections**    In a separate experiment, in the same setting, we vary the number of planning steps. As a learning method, we expect MGSC to be relatively insensitive to these variations. Uniform, in contrast, has no means to cope with an increase of erroneous model data.

Figure 6 shows the results. With a single query, MGSC and Uniform are effectively identical; there is little to distinguish their sampling distributions in this case. As the number of queries increase, Uniform exhibits declining performance. MGSC and Avoid Terminal remain robust. However, MGSC demonstrates superior performance to Avoid Terminal regardless of the number of queries.

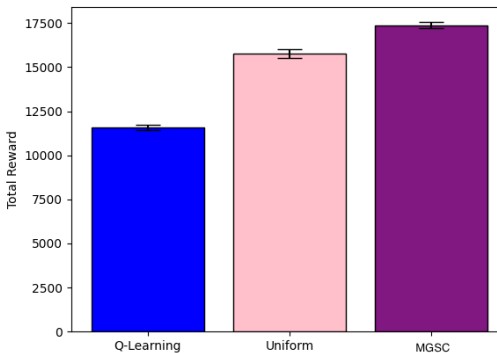 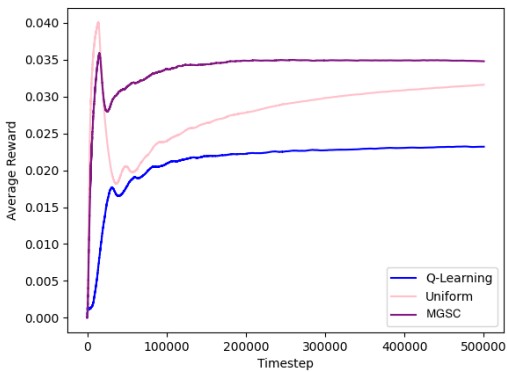

Figure 7: **TwoRooms Learned Model Performance**: (a) The total reward accumulated by each agent over the course of training. Error bars denote the 95% confidence interval. (b) The average reward accumulated during training for each agent.

## 4.2 TwoRooms Experiments

Our final experiment increases the difficulty of the learned model experiment by moving to a larger setting with more states. Figure11 shows the TwoRooms environment, a modification of the FourRooms environment introduced by Sutton et al. (1999). Goals cycle between the top and bottom right corners of the right room. The question this experiment asks is the same: can MGSC improve sample-efficiency when the model is flawed but continuously updates. However, the increased size of the environment means there are more states to select from when querying the model, including more states which are either irrelevant, or even detrimental, to efficient planning. Appendix B.2 describes the environment in more detail.

### 4.2.1 TwoRooms: Learned Model

Figure 7 shows the total and average reward achieved by Q-Learning, Uniform, and MGSC in the TwoRooms environment with a learned model. The conclusions from this figure are consistent with the results of the previous learned model experiment. MGSC achieves greater total reward than the Uniform agent, indicating that its learned search control distribution improved sample efficiency relative to this baseline.

A depiction of MGSC's learned distribution is shown in Figure 8 ~~in the Appendix~~. We observe that relatively little probability is applied to model states which produce erroneous rewards and states which are far from the shortest path to the goal. These are states in the upper-left corner of the left room, and states in the upper- and lower-left corners of the right room). We also observe a large amount of the probability mass assigned to state connecting the two rooms. This is surprising as its value does not change when the goal cycles from one state to another.

## 5 Summary and Future Work

Our paper studied the issue of sample-efficiency in reinforcement learning. We argued that search control was a promising avenue to further improvements, and that it was possible to learn search control strategies from experience. To support our argument, we introduced an algorithm (MGSC) that meta-learns a distribution over query states. The distribution was trained to improve planning-efficiency, and it was demonstrated, with empirical comparisons of total-reward, how MGSC increases sample-efficiency. Overall, we believe our results suggest useful directions for designing model-based RL systems that learn to perform search control.

We conclude by mentioning a few interesting directions of future work. Our study fixed the search control policy, $\tilde{\pi}$, to the behavior policy; are further improvements possible by learning a joint model of $\tilde{\pi}$ and $d$? Another line of questioning could focus on scaling. Specifically, what changes are necessary to support high-

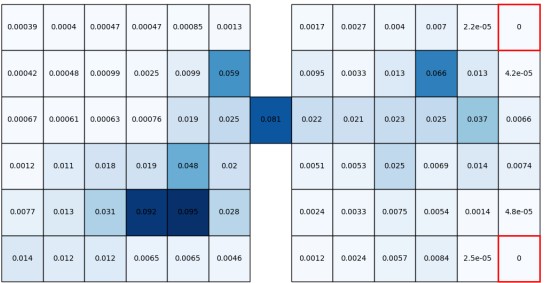

Figure 8: **TwoRooms Learned Model Solution**: The search control distribution learned by MGSC. As in Figures 3 and 5, numbers represent the probability mass of sampling a state from MGSC's TwoRooms model during planning. Greater probability mass corresponds to darker colors. Terminal states are outlined in red. As in other experiments, the search control distribution was initialized uniformly, the values pictured in this figure are from the end of training.

dimensional observations? Could the MGSC meta-loss (1) be calculated without enumerating over the entire state-action space? Current work that plans over discrete latent spaces could be relevant to this thread of research (Hafner et al., 2023).

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

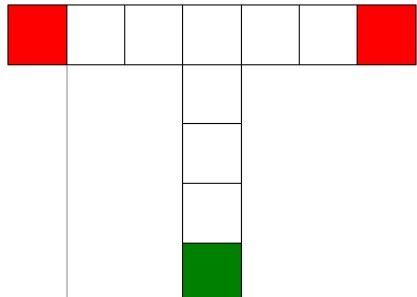

Figure 9: The TMaze environment. The green state indicates the agent's starting state while the red states indicate terminal states.

Erik Talvitie. Self-correcting models for model-based reinforcement learning. In *Proceedings of the AAAI Conference on Artificial Intelligence*, volume 31, 2017.

John N Tsitsiklis. Asynchronous stochastic approximation and q-learning. *Machine learning*, 16:185–202, 1994.

Christopher JCH Watkins and Peter Dayan. Q-learning. *Machine learning*, 8(3):279–292, 1992.

Stefan Radic Webster and Peter Flach. Risk sensitive model-based reinforcement learning using uncertainty guided planning. *arXiv preprint arXiv:2111.04972*, 2021.

David Wingate, Kevin D Seppi, and Sridhar Mahadevan. Prioritization methods for accelerating mdp solvers. *Journal of Machine Learning Research*, 6(5), 2005.

Peter R Wurman, Samuel Barrett, Kenta Kawamoto, James MacGlashan, Kaushik Subramanian, Thomas J Walsh, Roberto Capobianco, Alisa Devlic, Franziska Eckert, Florian Fuchs, et al. Outracing champion gran turismo drivers with deep reinforcement learning. *Nature*, 602(7896):223–228, 2022.

Annie Xie, James Harrison, and Chelsea Finn. Deep reinforcement learning amidst continual structured non-stationarity. In *International Conference on Machine Learning*, pp. 11393–11403. PMLR, 2021.

Zhongwen Xu, Hado P van Hasselt, and David Silver. Meta-gradient reinforcement learning. *Advances in neural information processing systems*, 31, 2018.

Tom Zahavy, Zhongwen Xu, Vivek Veeriah, Matteo Hessel, Junhyuk Oh, Hado P van Hasselt, David Silver, and Satinder Singh. A self-tuning actor-critic algorithm. *Advances in neural information processing systems*, 33:20913–20924, 2020.

# A  TMaze Experiments

## A.1  The TMaze Environment

We evaluate the MGSC algorithm in the TMaze; an episodic grid-world environment pictured in Figure 9. The TMaze is a non-stationary domain in which algorithms capable of adapting to a changing reward structure stand to perform well.

In the TMaze, an agent begins at a starting state and must navigate a vertical hallway, then turn left or right at a junction. Reaching a state at either the left or right of the horizontal hallway results in the termination of an episode. One of the terminal states emits a reward of +1 while the other emits 0. Every 600 episodes the rewards are swapped between terminal states. From the agent's perspective, the TMaze is thus non-Markov and non-stationary. At any timestep a random transition to an adjacent state may occur with probability $\epsilon_{\text{env}}$. A key element of the TMaze is that under the optimal policy only the values of certain states change. The values of states along the vertical hallway do not change when the reward is swapped, while the values of states in the horizontal hallway do change.

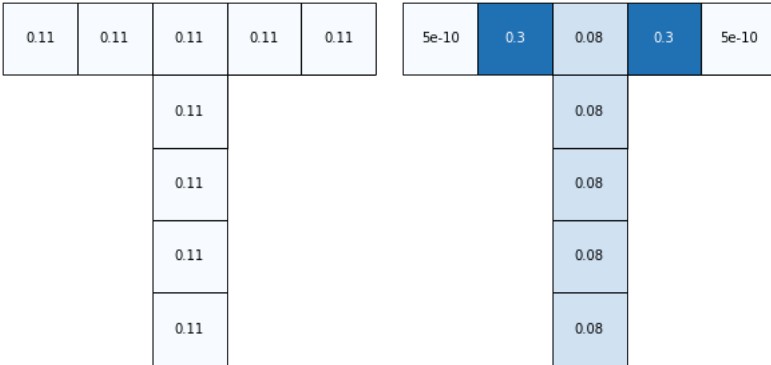

Figure 10: (a) The Uniform search control distribution. (b) The Avoid Terminal search control distribution. Terminal states are not pictured as no probability is assigned to these states. Darker colors indicate greater probability mass while text indicates the probability of sampling the corresponding state.

## A.2 Experimental Details

We describe some important experimental details useful in replicating the results of this work. In all experiments, each agent takes 250,000 steps in the TMaze environment. All agents used a discount of $\gamma = 0.9$. With the exception of the robustness to imperfections experiment, all planning agents perform 5 updates using transitions sampled from their model per environment interaction.

All results are computed by averaging over 30 different random seeds. Comparisons between different agents are always between the best hyperparameters for each agent. Additionally, visualizations of the search control distributions of Uniform and Avoid Terminal are pictured in Figure 10.

## A.3 Hyperparameter Selection

| Hyperparameter | Values |
|---|---|
| Step-size | 1e-3, 5e-3, 1e-2, 5e-2, 1e-1, 5e-1, 1e0 |
| Meta Step-size | 5e-5, 5e-4, 5e-3, 5e-2, 5e-1 |
| $\epsilon_{\text{policy}}$ | 1e-1 |
| $\epsilon_{env}$ | 1e-1 |

Table 1: Hyperparameters and values considered during grid search. Note that Meta Step-size is only used by the Meta Gradient Search Control Algorithm.

To select hyperparameters, we perform a grid search over all possible hyperparameter configurations from Table 1. Each configuration is run with 30 random seeds during the selection process. We average results from all seeds and report the results of the best hyperparameters for each algorithm in consideration.

# B TwoRooms Experiments

## B.1 The TwoRooms Environment

The TwoRooms environment is a non-stationary and stochastic gridworld domain. At the outset of each episode the agent begins in the bottom left corner of the domain. The agent must navigate a gridworld which is divided into two rooms with an opening through which the agent can pass. The agent's goal is to move from its starting position to a goal state. The agent may move in any of the four cardinal directions, receiving a reward of 0 after taking any action unless the agent reaches the goal state. Upon reaching the goal state, the agent receives a reward of +1 and the episode terminates. If the agent reaches a goal state

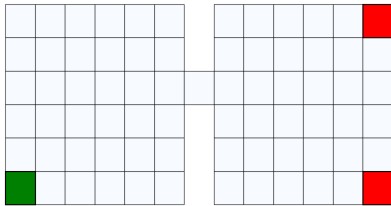

Figure 11: The TwoRooms environment. The agent's starting position is shown green. Possible goal positions are shown in red.

which is currently inactive, the episode terminates but the agent receives a reward of 0. With probability $\epsilon_{env} = 0.1$ the agent's action may fail, and a random action will be executed. Every 600 episodes, the position of the goal state is swapped.

## B.2 Experimental Details

We compare the performance of MGSC in TwoRooms against two baselines: Q-Learning and Uniform. These baselines are exactly analagous to the Q-Learning and Uniform agents described in prior sections. In these experiments, MGSC and Uniform are equipped with the learned model of the environment introduced earlier. That is, the model's dynamics exactly match the real environment, however, rewards are sampled proportionally to the count of each reward value observed thus far.

Experiments performed in this domain were run for a total of 500,000 timesteps. As in the TMaze results are averaged over 30 different random seeds. Results are reported for the best hyperparameter settings for each agent according the the total reward accumulated during training. The hyperparameter selection process was the same as that used in the TMaze experiments and considered the same possible values.

## B.3 Additional Experimental Results

Figure 8 shows the search control distribution learned by MGSC averaged over all random seeds. Notably, MGSC learns to place near-zero probability on states adjacent to goals states. As the learned model will converge to erroneous rewards over time, MGSC has learned that planning from these states is detrimental to value-function learning. Further, we observe that little probability is placed on states which are not along the shortest path to a goal state (e.g. states in the upper left corner of the left room, and states in the upper and lower left corners of the right room). This appears to show that MGSC has learned to avoid placing probability on states which are not important to explore in order to reach a goal state. We also observe that MGSC learns to place a large amount of probability mass on the state connecting the two rooms. This is surprising as the value of this state will not change when the goal cycles from one state to another.

# C An Extended Summary of Related Work

Our study builds on the insights of prior work in dynamic programming and RL. The first example comes from Tsitsiklis (1994), who proves that the convergence of a value function is independent of the ordering of transitions used for its update, provided they are experienced infinitely often. However, some orderings are better than others—as the work of Prioritized Sweeping demonstrates (Bertsekas & Tsitsiklis, 2015). Furthermore, these methods require a perfect model, which suggests that further research is needed before they can apply to settings where the model is learned.

Learned models introduce a number of complications that can interfere with priority estimation. For instance, learned models can lead to incorrect priority estimates when they predict the wrong outcomes. Consider a student that believes spending hours memorizing all the definitions in a dictionary will make them a great writer. This misinterpretation of the facts can result in them neglecting to practice their writing skills, which is actually the key to becoming a better writer. In other cases, inaccurate or irrelevant predictions made by models can worsen value estimates and result in similarly poor priority estimates.

Coping with imperfect models has become an active research area recently. Abbas et al. (2020) argues that epistemic uncertainty should guide the selection of model experience used for Dyna-style planning. This aligns with general wisdom that the agent should refrain from using the model where it is harmful. In a similar vein, Webster & Flach (2021) show how to balance epistemic and aleatoric uncertainty with reward penalties imposed on a model's output. Pan et al. (2020) take a different approach; they suggest that a model's states should be queried in proportion to the difficulty of learning an accurate value approximator—measured through the function's high-frequency content. Buckman et al. (2018) and Feinberg et al. (2018) adjust the planning horizon as a means to control for model error and value function bias. Learning progress is another important factor; the agent should not expend needless computation on states where the value has stabilized to a good estimate (Lopes et al., 2012). All of these approaches share the common goal of incorporating effective planning behavior as a bias in the learning system.

Ultimately the effectiveness of a particular bias depends on how well it aligns with the agent's overall goal to maximize reward (Lambert et al., 2020). Recent work has explored ways of aligning the model learning process with the agent's overall objective. In particular, Saleh et al. (2022) considers the problem of policy evaluation and proposes to train a model so that its output—including the query state—improves the credit assignment from planning. Value targeted regression (Ayoub et al., 2020) and the principle of value equivalence (Grimm et al., 2020; 2021; Arumugam & Van Roy, 2022) offer further ways to conceptualize alignment with the downstream control objective. Empirical evidence suggests that adopting such goal-oriented approaches can lead to improved sample-efficiency (Silver et al., 2017; Schrittwieser et al., 2020).

Finally, a natural consideration is to a learn bias for effective planning directly from environment interaction. To this end, our research draws inspiration from meta-gradient methods (Xu et al., 2018; Beck et al., 2023). Perhaps most closely related to our work is the approach of Flennerhag et al. (2022), who demonstrate improved sample-efficiency when adjusting typical hyperparameters such as the step-size and discount factor. Their method adapts the system's learning approach to improve downstream performance on the control objective. However, in contrast to their work, our aim is to adapt the distribution that determines where to plan with an imperfect model.

By prioritizing samples that minimize squared parameter error, MGSC prioritizes states where the value is inaccurate. Additionally, the loss function down-weights the priority of states whose values have sufficiently converged, since these states will not result in significant loss reductions. The loss function also assigns lower priority to states where the model is less accurate, as planning from these states could push the values further away from the optimal parameters.

