# OpenReview forum: "Meta-Gradient Search Control: A Method for Improving the Efficiency of Dyna-style Planning"
_TMLR — Rejected by TMLR_

### Review · Reviewer_tA8w · 2024-07-27

**Summary Of Contributions:**

This paper introduces an online meta-gradient algorithm for reinforcement learning (RL) from an imperfect model of the environment, specifically within planning algorithms such as Dyna. The authors claim that their method improves sample efficiency compared to conventional sampling strategies. The key contributions of the paper include:

- Development of a meta-gradient algorithm for search control in model-based RL.
- Demonstration of the method's efficiency in non-stationary environments where the dynamics change.
- Claims of the method's applicability to continual learning scenarios where environment dynamics change over time.

**Audience:**

Yes

**Broader Impact Concerns:**

I do not see any immediate concern about the impact of the paper.

**Claims And Evidence:**

Yes

**Requested Changes:**

- Provide a detailed explanation of how the samples' ability to improve the downstream planning process is determined.
- Include comparisons with established model-based RL methods and relevant baselines in the literature (e.g., Flennerhag et al. (2022)), as well as standard benchmarks (e.g., Gym, DeepMind Control Suite) to demonstrate the generalizability and effectiveness of the proposed method.
- Clearly explain all figures and evaluation metrics used in the paper, particularly Figures 3 and 5.

Recommended Changes:
- Conduct experiments in more diverse environments to show that the proposed method is not limited to the TMaze environment.
- Address the claim about handling imperfect models more convincingly by explaining how the method deals with models that do not converge to perfection.
- Justify the choice of changing environment dynamics every 600 episodes and discuss its implications for standard RL settings.
Provide more details on the baselines used (e.g., Avoid Terminal) and explain why they are appropriate for comparison.

**Strengths And Weaknesses:**

Strength:
- Addresses the important problem of changing environment dynamics, tackling a significant challenge in RL.
- Overall, the paper is easy to read, and it discusses relevant background information in many cases.
- Experimental results show that the proposed method achieves better sample efficiency compared to the baseline method evaluated.

Weakness:
- Several aspects of the proposed method, such as how the samples' ability is determined, are not clearly explained.
- The paper lacks comparisons with standard model-based RL methods. The baselines used (e.g., Uniform, Avoid Terminal) are not well-documented or justified.
- The experiments are limited to a specific TMaze environment, which raises concerns about the generalizability of the results to other RL settings.
- Several claims need further evidence.

Here are my detailed comments and feedback:

“..meta-learning algorithm that evaluates model queries based on the samples’ ability to improve efficiency of the downstream planning process”
how the samples' ability is determined?

Figures and evaluation metrics (e.g., Figure 3) are not clearly explained, making it difficult to understand the results and claims.
The claim 'The distribution MGSC learns (Figure 3)...it avoids states where the model is inaccurate (i.e., terminals) and updates are redundant (i.e., the vertical hallway), and it places more probability .. between goal switches (i.e., the horizontal hallway)' is not clear to me. How can this figure demonstrate the claim? I think better evidence is needed, or the claim should be toned down.

The paper's setting of changing environment dynamics every 600 episodes deviates from standard RL settings, making it less applicable to typical RL problems. What is the rationale behind this choice?

It is mentioned in the baseline, 'One model-based algorithm (Uniform) queries initial states …..' but there is no reference to it. 'Avoid Terminal' seems like an ad hoc approach that is used, and it is unclear if it can be applied in general settings.

I suggest evaluating some standard benchmarks such as Gym (Gymnasium), classic control, continuous tasks, and the DeepMind Control Suite, as existing model-based methods have been evaluated.

The paper mentions an 'imperfect' model, but eventually, the model is learned and used, so in principle, the model becomes perfect as the algorithm converges or the model learning component converges. Thus, how is this related to the paper's claim that it handles an imperfect model?

The setting of robustness to imperfection on page 7 is not clear. What does it mean to vary the number of planning steps? What was the number used for the experiments shown in Figures 2 and 4? It seems to be 5 steps, according to section B.2. What about using only 1 step in Figures 2 and 4, as that might be better for all the models? What would be the problem with doing that? How did you choose the update step of 5?

In Figure 7, what is the 'Adaptive' model? Is it the same as MGSC?

Overall, the paper presents an interesting approach to improving sample efficiency in model-based RL through a meta-gradient algorithm. However, the lack of clarity, insufficient baseline comparisons, and limited generalizability of the results warrant significant revisions before the paper can be considered for acceptance. Addressing the critical and recommended changes will strengthen the work and provide a more comprehensive evaluation of the proposed method.

---

> ### Author Response · Authors · 2024-08-17
> **Response to reviewer tA8w part 1**
>
> We thank the reviewer for their thoughtful comments. We have responded to your points below and provided more detail in the general rebuttal. Please let us know if our revisions helped and if any points need further clarification.
>
> > Include comparisons with established model-based RL methods and relevant baselines in the literature (e.g., Flennerhag et al. (2022)),
>
> The method introduced by Flennerhag et al. (2022) is a meta-learning method—not decidedly applicable to the model-based RL setting. Aspects of MGSC draw inspiration from that approach, but our algorithm is distinct and only applicable to search control. The baselines we have chosen are the only methods available in the domains we consider—given different amounts of prior knowledge.
>
> >  “..meta-learning algorithm that evaluates model queries based on the samples’ ability to improve efficiency of the downstream planning process” how the samples' ability is determined?
>
> Thank you for this question, we’ve adjusted this piece of text to make its intention clearer. MGSC’s meta-loss evaluates the benefit of performing a planning update from a sampled state, $s\sim d(\cdot;\eta)$ by approximating how well this update improved the agent’s value function parameters $\theta$. By adjusting $\eta$, and hence the search control distribution from which MGSC samples, according to parameter error, MGSC is able to improve the efficiency of its planning process.
>
> > Figures and evaluation metrics (e.g., Figure 3) are not clearly explained, making it difficult to understand the results and claims. The claim 'The distribution MGSC learns (Figure 3)...it avoids states where the model is inaccurate (i.e., terminals) and updates are redundant (i.e., the vertical hallway), and it places more probability .. between goal switches (i.e., the horizontal hallway)' is not clear to me. How can this figure demonstrate the claim? I think better evidence is needed, or the claim should be toned down.
>
> Thank you for this question. We have adjusted the text provided with Figure 3 to hopefully address the issues with clarity. The figure is meant to show MGSC’s learned search control distribution over the course of the experiment; with the numbers representing the probability at which a state is sampled during a planning step. This is then used to demonstrate MGSC’s ability to shift probability away from the terminal states which produce erroneous rewards and those states adjacent to them, and towards states which would more likely benefit MGSC by being sampled during planning.
>
> > The paper's setting of changing environment dynamics every 600 episodes deviates from standard RL settings, making it less applicable to typical RL problems. What is the rationale behind this choice?
>
> We chose the current value (600 episodes) to ensure all algorithms had sufficient data to achieve approximately asymptotic performance. Decreasing the frequency at which the goal state switches would not significantly alter the current results. However, if the frequency were to increase the environment would be far more challenging for all algorithms. In that scenario, the algorithms would not be given a sufficient amount of time to learn the dynamics of the environment before the reward regime swaps.

---

> ### Author Response · Authors · 2024-08-17
> **Response to reviewer tA8w part 2**
>
> > It is mentioned in the baseline, 'One model-based algorithm (Uniform) queries initial states …..' but there is no reference to it. 'Avoid Terminal' seems like an ad hoc approach that is used, and it is unclear if it can be applied in general settings.
>
> The use of the term “Uniform” was not intended to imply a previously named algorithm. The pseudocode matches the newly-added pseudocode for the Dyna architecture (Algorithm 1), with the distribution at which (s,a) are selected from the model being uniformly random. The “Avoid Terminal” algorithm was chosen as a baseline to represent an algorithm designed with expert knowledge of the environment.
>
> > The paper mentions an 'imperfect' model, but eventually, the model is learned and used, so in principle, the model becomes perfect as the algorithm converges or the model learning component converges. Thus, how is this related to the paper's claim that it handles an imperfect model?
>
> In this case, the model is imperfect because it actually does not converge to optimality. The model returns rewards from its terminal states in proportion to the count of rewards that have been observed in the real environment. The environment periodically switches between reward regimes. This means that in the limit the model will have observed equal counts of 0 and 1 at both terminal states. Thus, the model will output 0 or 1 with equal probability regardless of which terminal state in the real model is currently returning a reward of 1. As such, even in the limit the model remains imperfect as it does not match the reward dynamics of the real environment.
>
> > The setting of robustness to imperfection on page 7 is not clear. What does it mean to vary the number of planning steps?
>
> Thank you for this question. Hopefully by addressing the confusion regarding Dyna-style algorithms we have addressed the clarity issue regarding the number of planning steps.
>
> >What was the number used for the experiments shown in Figures 2 and 4? It seems to be 5 steps, according to section B.2.
>
> You're correct, Figures 2 and 4 use 5 steps.
>
> >What about using only 1 step in Figures 2 and 4, as that might be better for all the models? What would be the problem with doing that? How did you choose the update step of 5?
>
> Decreasing the number of planning steps for the model-based approaches would not serve to highlight the relevance and necessity of using a model in the chosen environments – which is the area our work hopes to address. Increasing the number of planning steps given to the model-based algorithms would only serve to increase their performance and learn the dynamics of the environment and the sample distribution of MGSC faster. We had run experiments varying the number of steps to 10 and 20, and the results did not significantly change in such a way that the evidence for the claims made in the paper were any different. As such, 5 is chosen as a balanced value between being able to appropriately show off the benefit of a model in the chosen environments and the computational cost of doing more updates as a result of a higher number of planning steps.
>
> > In Figure 7, what is the 'Adaptive' model? Is it the same as MGSC?
>
> Thank you for pointing this typo out. This has now been fixed and relabeled in the figure as MGSC.

---

### Review · Reviewer_wkAx · 2024-08-03

**Summary Of Contributions:**

This paper proposes a meta-gradient algorithm that adjusts the probability with which states are queried during Dyna-style planning. However, the concept and implementation details are not clearly articulated, making it difficult to grasp the core contributions. The experiments are conducted on Tmaze and TwoRooms environments, demonstrating the improved sample efficiency of the proposed method.

**Audience:**

Yes

**Claims And Evidence:**

No

**Requested Changes:**

See Above

**Strengths And Weaknesses:**

**Strenghts**

- The approach addresses an important aspect of reinforcement learning, focusing on improving sample efficiency.

**Weaknesses**

- The paper is extremely difficult to follow due to unclear explanations and vague terminology. For example, the term "knowledge" in the third paragraph of page 1 is ambiguous. Does it refer to the model's understanding of the environment, or something else? Similarly, terms like "trustworthy" and "imperfections" need precise definitions to avoid confusion.

- Dyna-Style Planning: The paper does not provide a clear definition or explanation of Dyna-style planning. This is crucial as it is not a standard concept universally understood in the literature. A self-contained description would significantly improve comprehension.

- Meta-Gradients: The concept of meta-gradients is introduced without adequate context or explanation. The term "Meta" remains unexplained even after a thorough read of the paper. It is essential to clarify this to ensure the readers understand the innovation.

- The role or meaning of $\eta$ is unclear. While it is mentioned as representing the initial state distribution, its use in section 3.1 suggests a generative model or state distribution generated by some behavioral policy. This inconsistency needs to be addressed for better understanding.

- Appendix A: Empty section.

Overall, the paper requires substantial revisions to improve its readability and coherence. A thorough rewrite is necessary to achieve the clarity needed for a fair evaluation.

---

> ### Author Response · Authors · 2024-08-17
> **Response to reviewer wkAx**
>
> We thank the reviewer for their thoughtful comments. We have responded to your points below and provided more detail in the general rebuttal. Please let us know if our revisions helped and if any points need further clarification.
>
> On the use of “knowledge”: We have updated two instances of this word to “value estimate”.
>
> On the use of “trustworthy”: We intended this term to imply its everyday, common-use meaning as sufficiently accurate to be trusted.
>
> On the use of “imperfections”: with this term our intention was to describe the holistic state of inaccuracy due to multiple sources.
>
> On the role of $\eta$: Thank you for pointing this out. We realize that using the term “initial state distribution” for the search control distribution made their role unclear. We have removed this text to reduce confusion. In general, eta is a real-valued vector that parametrizes the search-control distribution. We introduce and define these in Section 2.3. They are also the meta-parameters MGSC tunes. This is described in Section 3 and Algorithm 2.

---

### Review · Reviewer_QJcf · 2024-08-11

**Summary Of Contributions:**

This work studies the problem of planning when the environment model is inaccurate. A meta-gradient algorithm is proposed to address this unreliability issue: it optimizes the distribution according to which model queries are performed, a method designated as search control. In practice, the authors aim to optimize a parametrized distribution $d_{\eta}\in\Delta_{\St}$ on top of a Dyna-style algorithm. To do so, they define a meta-loss function directing the $\eta$ parameters towards distributions that most positively affect the Q-value parameter updates. The proposed algorithm, MGSC, shows improved sample efficiency on non-stationary domains.

**Audience:**

Yes

**Claims And Evidence:**

Yes

**Requested Changes:**

To secure my recommendation for acceptance, I suggest the authors broaden the benefits of their approach compared to other methods in non-stationary environments, extend MGCS to large internal states, or at least test the current method to a larger set of environments.

**Strengths And Weaknesses:**

*Strengths*
The paper is very well-written. It gives a good motivation to the proposed algorithm and the conducted experiments support the paper's objectives.

*Weaknesses*
I am not familiar with the literature on Dyna-style algorithms. However, other methods exist to address MDP non-stationarity using model-based RL: these rely on change-point detection [1], variational intference [2], or other structural assumptions on the MDP changes [3]. This work could broaden its impact by relating and comparing itself to these alternative methods.

Although some techniques are proposed as an extension to large internal state-spaces, these are missing in the experiments. I think they would strengthen the impact of this work.

The approximate target defined in Sec 3.1 relies on a one-step TD error, which could be insufficient when the Q-value is too low/the current policy is bad. I may be missing something here but if not, it would be interesting to see the effect of eligibility traces on training.


[1] Alegre, Lucas N., Ana LC Bazzan, and Bruno C. da Silva. "Minimum-delay adaptation in non-stationary reinforcement learning via online high-confidence change-point detection." arXiv preprint arXiv:2105.09452 (2021).
[2] Xie, Annie, James Harrison, and Chelsea Finn. "Deep reinforcement learning amidst continual structured non-stationarity." International Conference on Machine Learning. PMLR, 2021.
[3] Chen, Xiaoyu, et al. "An adaptive deep rl method for non-stationary environments with piecewise stable context." Advances in Neural Information Processing Systems 35 (2022): 35449-35461.

---

> ### Author Response · Authors · 2024-08-17
> **Response to reviewer QJcf**
>
> We thank the reviewer for their thoughtful comments. We have responded to your points below and provided more detail in the general rebuttal. Please let us know if our revisions helped and if any points need further clarification.
>
> On Dyna-style algorithms: To clarify the ambiguity around Dyna-style algorithms, we have added an additional section on Dyna-style algorithms along with pseudocode.
>
> On non-stationary in MBRL: Thank you for pointing out additional work related to non-stationary MDPs in MBRL. We have included additional citations in our section on learning models to acknowledge these works.
>
> On large internal state-spaces: please see the comment in our general response. We hope this helps to address this particular concern.
>
> On the approximate target definition: This is a great point. In an earlier iteration of this work, we experimented with using multiple steps of TD updates to construct the approximate target. However, we found empirically better performance for our algorithm using only the one-step TD update. We also considered the use of eligibility traces, but haven’t yet had the opportunity to investigate this. However, we agree that revisiting methods for constructing the approximate target would be a fruitful direction for extensions of our project.

---

### Author Response · Authors · 2024-08-17
**General Response**

Thank you to all the reviewers for your thoughtful comments. We were glad to see that both reviewers QJcf and tA8w found our paper “very well-written” and “easy to read.” We have revised our manuscript based on the feedback and marked the edits with red text.

Several reviewers expressed concern about the scale of the TMaze domain and whether our method can extend to domains with larger state spaces. Note, our submission included results from a larger gridworld, the TwoRooms domain. However, some of this information was placed in the Appendix, so it may have gone unnoticed. We have therefore brought that content into the main body of the revised manuscript. In this domain, we observe that MGSC improves the sample efficiency of the underlying learning process.

To the same end, there was a request for our experiments to include Gym and the DM Control Suite. Although MGSC is theoretically extensible to problems requiring function approximation, the method used in our experiments does not directly apply to continuous settings like Gym and the DM Control Suite. This method in our study was chosen as an empirical device—to demonstrate that search-control can be meta-learned. This is an important first step toward learning to perform search control in the full function approximation setting, which we leave for future work.

Do our results from the TwoRooms domain sufficiently demonstrate that MGSC is not restricted to the TMaze?

Some other general points of concern are as follows.
On Dyna-style algorithms: Dyna-style algorithms are a class of model-based RL algorithms where planning is employed for credit assignment—whether to update a value function or a policy. These stand in contrast to the class of Decision-time Planning algorithms where planning is used to select actions, e.g. Model-Predictive Control and AlphaGo.

In our paper, we provided more background information on Dyna-style algorithms, including pseudocode, more references, and more details in our description of the algorithm.

Meta-learning background: We added a new Meta-learning section into our Background. This introduces the concept of meta-learning and it cites relevant work on the meta-gradient technique, which our paper adapts to model-based RL.

---

### Author Response · Authors · 2024-08-30
**Further questions or clarifications?**

Thanks again to the reviewers for their thoughtful questions. Following our responses and the changes we've made to the manuscript, we're wondering if the reviewers have any additional questions or clarifications?

Specifically, we also wanted to ask whether moving the second experiment, in the TwoRooms environment, from the appendix to the main body of the paper helped to address concerns that the results might be limited to the TMaze domain?

---

### Decision · Action_Editor_pUGW · 2024-10-31

**Recommendation:** Reject

**Comment:**

We first note that the comments about the lack of background regarding the definition of dyna-style algorithms have been adequately addressed by the authors in the rebuttal.  The recommendation to reject the paper is based on the fact that the proposed technique is limited to discrete states, the experiments consist of a limited proof of concept and the is a lack of theory.  The proposed meta-learning technique is very interesting and promising.  However, it is not clear to what extent it is a heuristic with limitations/pitfalls or a principled approach.  This is where a theoretical analysis with more extensive experiments would be desired.  The fact that the two testbeds are discrete toy domains (i.e., grid worlds) is not helping.  There is no comparison with other techniques that can deal with non-stationary environments in model-based RL.  There is no demonstration that the proposed technique can advance the state of the art in real-world applications.

Nevertheless, the research is good. The authors are encouraged to continue their work and possibly submit a major revision.

**Audience:**

This work will be of interest to the RL community.

**Claims And Evidence:**

The paper presents a new meta learning technique to guide the search in dyna-style model based RL with application to non-stationary domains.  The effectiveness of the proposed meta learning technique is demonstrated empirically in two toy domains with discrete states.  There is no theoretical analysis.  Overall, the evidence is insufficient.

**Resubmission Of Major Revision:**

The authors may consider submitting a major revision at a later time.